# Influence of Graphene Sheets on Compaction and Sintering Properties of Nano-Zirconia Ceramics

**DOI:** 10.3390/ma15207342

**Published:** 2022-10-20

**Authors:** Elena A. Trusova, Dmitrii D. Titov, Asya M. Afzal, Sergey S. Abramchuk

**Affiliations:** 1Baikov Institute of Metallurgy and Materials Science, Russian Academy of Sciences, 49 Leninsky pr., 119334 Moscow, Russia; 2Ceramic Materials Engineering, University of Bayreuth, Ludwig-Thima-Ste. 36b, 95447 Bayreuth, Germany; 3Faculty of chemistry, Lomonosov Moscow State University, GSP-1, Leninskie Gory, 119991 Moscow, Russia

**Keywords:** graphene-zirconia, nano-zirconia, rheology of nanostructures, activation energy of the sintering, sonochemistry

## Abstract

The use of a nanostructured graphene-zirconia composite will allow the development of new materials with improved performance properties and a high functionality. This work covers a stepwise study related to the creation of a nanostructured composite based on ZrO_2_ and graphene. A composite was prepared using two suspensions: nano-zirconia obtained by sol-gel synthesis and oxygen-free graphene obtained sonochemically. The morphology of oxygen-free graphene sheets, phase composition and the morphology of a zirconia powder, and the morphology of the synthesized composite were studied. The effect of the graphene sheets on the rheological and sintering properties of a nanostructured zirconia-based composite powder has been studied. It has been found that graphene sheets in a hybrid nanostructure make it difficult to press at the elastic deformation stage, and the composite passes into the plastic region at a lower pressure than a single nano-zirconia. A sintering mechanism was proposed for a composite with a graphene content of 0.635 wt%, in which graphene is an important factor affecting the process mechanism. It has been determined that the activation energy of the composite sintering is more than two times higher than for a single nano-zirconia. Apparently, due to the van der Waals interaction, the graphene sheets partially stabilize the zirconia and prevent the disordering of the surface monolayers of its nanocrystals and premelting prior to the sintering. This leads to an increase in the activation energy of the composite sintering, and its sintering occurs, according to a mixed mechanism, in which the grain boundary diffusion predominates, in contrast to the single nano-zirconia sintering, which occurs through a viscous flow.

## 1. Introduction

In the last decade, the growing interest of the developers of new materials has been turned to graphene-ceramic hybrids in particular, based on zirconia, in whose structure the graphene particles or its derivatives (reduced oxidized graphene or oxidized graphene) are incorporated, since they have unique physicochemical properties [1]. The spectrum of technologies (incomplete) interested in such a row of materials includes: dual technologies, the development of small-sized electronic devices, photonics, the creation of miniature energy sources and batteries, the production of medical preparations for the vector delivery of drugs in the body and early diagnosis of diseases (at the cellular level), the development of implants and antimicrobial reagents, the creation of nanostructured (photo)catalysts for petrochemical processes and environmental protection, and in the many other industries [2,3,4,5,6,7]. However, despite the high demand, on the one hand, and a considerable number of publications, on the other hand, the studies known from the open scientific press are scattered, and, as a rule, narrowly focused; they rarely affect the study of the mechanisms of structure formation at the molecular level. It turned out to be difficult to implement in practice, the creation of a reliable method for the synthesis of hybrids with a controlled structure and properties [3,8,9,10]. Therefore, the selection of the optimal process modes for obtaining nanostructured powders and fine-grained ceramics, is most often carried out empirically and is intended to solve a specific practical problem [11,12]. Thus, one of the main problems in synthesis composites is ensuring the uniform distribution of graphene in the materials.

Of the three technologically acceptable graphene species, only pure, that is oxygen-free graphene can significantly improve the performance of hybrid ceramic materials, since only it has a defect-free sp^2^-electron system that provides unique electronic properties. Compared to the oxygen-free graphene, the oxidized graphene and the reduced oxidized graphene showed only a slight increase in the fracture toughness through their reduced mechanical strength and structural defects [13]. It is shown that the addition of graphene to Zr-ceramic, in no more than a 1 vol%, makes it possible to obtain a composite material with an increased density, hardness, and wear resistance [14]. The inclusion of zirconia nanoparticles in the graphene sheets leads to the creation of promising material for lithium-ion battery anodes [15]. Such a graphene-zirconia composite showed a high specific electrical capacitance, excellent speed performance, and a high cycling stability. Through the stable zirconium oxide sandwich structure, the stable construction supports the graphene, which provides more active sites for the Li^+^ incorporation. It has been determined that graphene in a nanolaminated structure, can significantly increase the composites’ hardness and strength without reducing the plasticity, which underlines the importance of the structural design using it.

D. Guo et al. [16] proposed a reasonable and simple method for the synthesis of composite aerogels consisting of reduced oxidized graphene and mesoporous zirconia. Epichlorohydrin and dimethylformamide have also been used to reduce oxidized graphene in the synthesis. Du et al., described the preparation, characterization, and electrochemical properties of the graphene-zirconia nanocomposites and their application for the enrichment and detection of methyl parathion [17]. However, none of the mentioned composites was in the condensed state of an aerogel and thus is limited to the applications in many areas, due to the rather low specific surface area of such composites.

A few years ago, the methods were developed for the deposition of zirconia on oxidized graphene [18,19,20]. M. Liang et al. [21] considers the graphene action on the structure and the characteristics of the electrodes; it summarizes the latest advances in the field of graphene-containing anode materials. Several methods have been developed for incorporating zirconia into graphene-containing materials. Since the mechanical mixing of carbon material with metal oxides does not lead to the formation of a hybrid material with a single phase, the methods using colloids, solid-state pyrolysis, and chemical interactions were considered as an alternative. These methods lead to the decorated and mesoporous systems products, core-shell particles, etc. The nanosheets of graphene obtained by the graphite oxidative exfoliation and subsequent chemical recovery, were chosen by E. Yoo et al., as the material of the anode [22]. M.R. Shaik et al. [23] considers a solvothermal approach to the preparation of the stabilized nanoparticles of zirconium oxide, a highly reduced graphene oxide, and a nanocomposite based on zirconia and a reduced graphene oxide solved in benzyl alcohol and a stabilizing ligand. The electronic interactions between zirconia and a highly reduced graphene oxide, have been proven by a comparative electrochemical study of pure zirconia and a highly reduced graphene oxide using cyclic voltammetry. H. Mudila et al., studied the influence of the graphene oxide content on the electrochemical properties of the composite. They found that an increase in the mass ratio of zirconia/oxidized graphene above 0.5, leads to a reduction in the specific electrical capacity of the composite [24]. Therefore, it has been suggested that the improvement in the electrochemical properties is due to the nanosized components and the increased electroconductivity of the graphene oxide. All of these factors enhance the pseudo capacitance in the charge transfer reaction of the nanocomposite, due to the fast and reversible redox reactions on its surface.

The difficulties of creating an electrode material with a stable structure have not yet been overcome. There is also a problem of of synthesizing high-quality graphene without oxygen-containing functional groups and with a low degree of agglomeration. At the same time, the powders’ rheological properties require special study. It was shown that the 0.05 wt% sheets’ presence of oxidized graphene in cement leads to an increase in the bending and compressive strength of the material by 15–33 and 41–59%, respectively [25,26,27]. Despite a large number of publications, there are practically none on the research of composites, based on oxygen-free graphene and metal oxides that would be acceptable for technological implementation [28,29]. A small number of works of this nature include the work by L. Dong [28], where the authors obtained a surprisingly highly concentrated graphene suspension (50 mg mL^−1^), at a pH value of the liquid medium, equal to 14 (harsh conditions) and our earlier work, in which milder conditions were used [30]. Therefore, the study of the sintering conditions for graphene-ceramic nanostructured powders is timely and in demand by the developers of new functional materials. Monolithic ceramic materials sintered from graphene-ZrO_2_ powders are less studied. However, the successful compaction and the sintering of the nanostructured composite powders requires completely new technological approaches, based on the results of the rheological studies and careful selection of temperature increase modes during the compaction and sintering. Among the few works of recent years, one can note the publication of R. Cano-Crespo et al., where the mechanical properties of composites based on ZrO_2,_ reinforced with oxidized graphene or carbon fibers were studied. The results obtained did not allow us to draw conclusions about the positive effect of carbon additives on the mechanical properties. However, it is noted that both the oxidized graphene and the carbon fibers cause a “lubricating effect”, which consists of increasing the mobility of the grain boundaries, i.e., the deterioration of the creep of the material. Apparently, in the experiment, the carbon-containing layers were too thick and had a poor adhesion to the ZrO_2_ grains. In addition, the increased distance between the carbon layers in the oxidized graphene leads to a weakening of the interlayer interaction, which, in turn, also contributes to the deterioration of the creep of the material [31]. S. Sagadevan et al., made an attempt to improve the potential properties of the powder composite, based on ZrO_2,_ by introducing not oxidized, but reduced oxidized graphene. The use of the hydrothermal method ensured the uniform deposition of the graphene on the surface of the ZrO_2_ particles and their almost complete coverage with a good adhesion [32].

Earlier, we published an article about a graphene-zirconia composite, which was synthesized from a Zr-containing sol and an oxygen-free graphene suspension, the essence of which was that the nano-zirconia crystallization occurred on the graphene sheets [33]. It was found that when the content of graphene in the composite is less than 1 wt%, its compressibility is significantly lower than that of single nano-zirconia obtained from the same sol. Apparently, this is due to the low elasticity of the oxygen-free graphene sheets, in which zirconia nanocrystals are incorporated.

We turned to the rheological studies, because the behavior of the systems we develop under the standard conditions, differs significantly from the behavior of the traditional submicron powders, due to the presence of the ZrO_2_ nanocrystals and the graphene sheets in the material, which significantly change the fluidity and compressibility of the powders. At the same time, there are practically no works covering these physicochemical characteristics.

We have determined the significant differences in the morphology of the composites obtained from the Zr-containing sol and the nanocrystalline zirconia. Therefore, in order to define the most optimal modes of compaction and the sintering of ceramics, it is important to determine the similarities and differences in the rheology and the sintering of the composites obtained by different methods. The present paper generalizes the results of a study covering the preparation of nano-zirconia, an oxygen-free graphene suspension and a composite, a comparison of the rheological properties and the sinterability of the pure zirconia and a composite, and also proposes a mechanism for sintering the composite, which reflects the decisive role of the graphene sheets. The use of oxygen-free graphene to obtain the composites, based on ZrO_2,_ will make it possible to avoid the disadvantages that the analogues containing oxidized and reduced oxidized graphene suffer from. This manuscript is practically the first where an attempt was made not only to accumulate the experimental facts, but also to give an explanation for them, based on the physicochemical features of the oxygen-free graphene.

## 2. Materials and Methods

### 2.1. Synthesis

#### 2.1.1. Preparation of the Zirconia Nanopowder

To obtain the zirconia nanopowder, a weighed portion of zirconyl chloride octahydrate ZrOCl_2_·8H_2_O (TU 71-085-39-2001, ChimMed, Moscow, Russia) was dissolved in deionized water with stirring on a magnetic stirrer (500 rpm) and with heating to 85–90 °C. A 0.1M aqueous solution of ZrOCl_2_ was used, to which monoethanolamine (MEA) (TU 2632-094-44493179-04, ECOS-1, Moscow, Russia) was added to form and stabilize the sol at a molar ratio of MEA/Zr = 1.5. At the beginning of the synthesis, the pH of the reaction mixture was 8.00–9.00 (±0.05). Then, the sol was subjected to evaporation at a temperature of 95–98 °C and stirred on a magnetic stirrer (300–500 rpm) until a gel was formed, followed by the heat treatment in an oven at 500 °C for 1 h.

#### 2.1.2. Obtaining a Graphene Suspension in Isopropanol

The powder of the synthetic graphite (NPO Unihimtek, Podolsk, Russia) with a 600–800 µm particle size, was used as a carbon source; the content of the trace amounts of sulfur and chloride ions did not exceed 10 ppm. The graphite powder was mixed with isopropanol; the molar ratio of graphite/isopropanol was 1/26. The resulting substance was subjected to an ultrasonic treatment in a Sonoswiss SW1H unit (power 200 W) for 2 h. The graphene suspension was separated from the unreacted graphite after the sedimentation for 20–22 h, by decantation of the light fraction.

#### 2.1.3. Preparation of the Nanostructured Composite Powder

In order to obtain a graphene-zirconia composite, a water-alcohol suspension of a nanocrystalline zirconia powder, calcined at 500 °C, was mixed with 800 mL of the graphene suspension obtained by the ultrasonic exfoliation in isopropanol for 2 h. The mixed colloid was obtained by heating to 60–65 °C and stirring (500 rpm) on a magnetic stirrer for 20–25 min, followed by the evaporation at a temperature of 95–98 °C, and stirred until a paste was formed, which was then transferred to a porcelain cup and placed in an oven, where the heat treatment was carried out in the air for 1 h at 400 °C.

### 2.2. Physicochemical Characterization

The elemental analysis of the nanopowders was carried out with a Perkin Elmer Optima 5300DV Inductively Coupled Plasma Optical Emission Spectrometer (Waltham, MA, USA) and Carbon/Sulfur Analyzer LECO SC-400. The surface and porosity of the powders were studied with N_2_ adsorption-desorption isotherms, using a Micromeritics TriStar 3000 and a NOVA 2200 (Norcross, GA, USA) specific surface analyzer. The Brunauer–Emmett–Teller (BET) method was used to determine the specific surface.

The morphology of the received powders and the graphene was explored by transmission electron microscopy (TEM) and electron diffraction using a LEO-912 AB OMEGA instrument at 100 kV, in addition to the high-resolution transmission electron microscopy (HRTEM, JEM 2010, JEOL Ltd., Tokyo, Japan) with the EELS attachment (GIF Quantum, Gatan Inc., Pleasanton, CA, USA). The Raman is with a laser (DXR™ 2 Raman Microscope, Thermo Fisher Scientific, Waltham, MA, USA) with a wavelength of 532 nm and a power of 6.0 mW. The accumulation time for each Raman spectrum was about 10 s.

The particle size distribution of the powder was detected using an automated diffusion aerosol spectrometer, Model 2702, Aeronanotech, Russia (DAS). The sample weight was 0.001 g; the objects was treated ultrasonically for 3 min. The relative error of measurement was 5%.

The graphene-zirconia and the single zirconia powders were studied by X-ray (DRON-3M, Russia, CuK_a_) and using JCPDS cards for the interpretation [34,35]. The studies were performed at room temperature and at a normal atmospheric pressure. The average crystallite size was calculated with the Rietveld method, viz. using an iterative procedure to minimize the experimental diffraction pattern deviations from those calculated.

### 2.3. The Study of the Rheological Properties and the Sintering Properties of the Synthesized Nanostructured Powders

#### 2.3.1. Rheological Measurements

The rheological measurement was performed on a mechanical testing machine (Instron 5581, High Wycombe, UK) at a constant loading speed of 1 mm/min. A weight of the testing objects powder in a steel mold was 0.15 g. Each sample of the composite was tested five times, then the values were averaged (the average value of the standard deviation was δ_1_ = 0.0744 for the graphene/ZrO_2_ and δ_2_ = 0.0476 for the single ZrO_2_).

The deformation was calculated using the method described in our previous study [33], in this case, the synthesized powders’ bulk density, after shaking, was determined by the Scott technique conformity with ASTM B329-18 [36].

#### 2.3.2. Dilatometric Study

The synthesized nanostructured powders’ dilatometry was conducted in a DIL 402 C Netzsch dilatometer (Netzsch, Germany),, according to the previously studied method [37]. The cylindrical green bodies obtained after the rheological test, with a diameter of 5 mm and a height ∼2.5 mm, were used for the dilatometry study. The thermocouple (tungsten-rhenium alloy) was located near the sample and its temperature was accurately recorded; the second thermocouple (tungsten-rhenium alloy) was in the chamber with the heater. This chamber has an argon atmosphere independent of the working chamber. The argon flow introduced into the furnace was 70 mL min^−1^, the heating rate was 5, 10 and 20 °C/min, and the heating continued to a temperature of 1750 °C, then cooling at the rate of 20 °C/min.

### 2.4. Calculation of the Parameters of the Sintering Process

The sintered ceramic density (ρ_s_) was calculated by the Equation (1):(1)$$\rho_{s}={ [\frac{1}{1+\frac{\Delta L}{L_{0}}-\alpha \left(T-T_{0}\right)}]}^{3}\cdot \rho_{g}$$

L_0_—sample initial length, ∆L/L_0_—relative linear shrinkage, T_0_—initial temperature, T—measured temperature, ρ_g_—green body density, α—coefficient of the thermal expansion (CTE).

The average value of the CTE was determined from the curve from the cooling after the sintering. The estimated calculations of the activation energy (Q) and the apparent activation energy (nQ, where n is the process order corresponding to its mechanism) of the sintering were carried out in the manner we previously discussed [38]. For the heating rate of 5°, 10°, and 20 °C/min, the graphs were plotted in the coordinates 1/T—ln [TC(dρ/dT)], and the tangent of slope S_1_ was defined. The activation energy value was calculated, according to Equation (2):Q = -R·S_1_(2)
where R is the universal gas constant.

The same heating rate graphs in the coordinates 1/T—ln [T(d∆L/L_0_)/dT] were constructed graphs, the tangent of the angle of which S_2_ makes it possible to determine the apparent activation energy of the sintering by the Equation (3):nQ = -R·S_2_(3)

Further, the order values of the sintering process for the pure nano-zirconia and the graphene-zirconia composite, were calculated by the Equation (4):n = nQ/Q = S_1_/S_2_(4)

## 3. Results and Discussion

The synthesis of the graphene-zirconia nanostructured powders consisted of three stages: exfoliation of the oxygen-free graphene sheets using the sonochemical method and the stabilization of its suspension (i), preparation of the ZrO_2_ nanopowder (ii), the obtaining of the hybrid nanostructures using the graphene suspension and the nanocrystalline ZrO_2_ powder (iii). Figure 1 shows the appearance of a graphene suspension in isopropanol after 24 h sedimentation and the repeated decantation. The bright Tyndall cone indicates a rather dense structure of the colloid formed by the suspended solid particles.

According to the Raman shown in Figure 2, all of the bands characteristic of the oxygen-free graphene, are present in the obtained spectrum. In the region of 1567 cm^−1^, the main spectral characteristic is observed in the form of a narrow band G, which arises as a result of the stretching vibrations in the plane of the carbon atoms bound by the sp^2^ bonds. The small width of the G band indicates the predominance of a low-layer, low-defect oxygen-free graphene. The latter is also confirmed by the EELS analysis data (inset): the absence of the 532 eV peak, characteristic of the oxidized graphene indicates the absence of the O-containing functional groups on the surface of the graphene sheets. At the same time, a broad peak centered at 284 eV, indicates the transition from 1s to π* and confirms the presence of the sp^2^ carbon atoms in the system. The D band in the region of 1347 cm^−1^, characterizes the degree of disorder in the electronic structure of the carbon layers, which occurs when the lattice moves from the middle of the Brillouin zone. A large value of intensity ratio I_G_/I_D_ indicates a low degree of defectiveness of the graphene. The 2D mode in the region of 2711 cm^−1^, is an overtone that characterizes the internal properties of the graphene and appears as a result of the double resonant Raman scattering. The overtone G* in the region of 2441 cm^−1^, is also characteristic of the graphene and is due to the presence of a few defects since the peak intensity is low [39,40].

According to the TEM data, the graphene in the suspension, consisted of submicron blocks sheets (Figure 3a). The electron diffraction indicates a set of differently oriented graphene sheets with multiple folds (Figure 3b). The sheet thickness did not exceed 3 nm (Figure 3c). The dark-field image in Figure 3d, shows that the graphene agglomerates consist mainly of one-two-layer sheets, as evidenced by the moiré pattern.

The phase composition and morphology of the zirconia nanopowder obtained by the sol-gel synthesis, were studied by XRD, TEM, electron diffraction, and N_2_ adsorption–desorption methods. The residual carbon content in the prepared single zirconia powder was 0.07 (±0.01) wt%.

According to XRD data (Figure 4), the zirconium oxide consisted of 46 wt% from the ZrO_2_ tetragonal modification tP6 (JCPDS card no. 24-1164) and 54 wt% from the ZrO_2_ monoclinic mP12 (JCPDS card no. 05-0543). The average size of the zirconia crystallites, calculated using the Rietveld method, was 9 nm for the monoclinic modification and 10 nm—for the tetragonal modification. According to TEM data, the zirconia powder consisted of agglomerates with sizes of 100–300 nm formed by 7–13 nm crystallites (Figure 5a), which are consistent with the calculation results from the XRD data. The dark-field images (Figure 5a,b) show that the powder was composed of well-crystallized discrete particles. The agglomerates composed of the nanocrystals have a mesoporous structure with pore sizes of 3–4 nm (Figure 5c).

The study of the synthesized zirconia nanopowder by the N_2_ adsorption–desorption method (Figure 6), indicates that it has a mesoporous structure; the hysteresis loop shown (inset in Figure 6), belongs to type IV (according to the IUPAC classification). The pore size was 3–4 nm, which was consistent with the TEM data (Figure 5c). The specific surface of the nano-zirconia powder, calculated using the BET method, was 53 m^2^/g. According to the DAS data, the average particle size in the obtained powders was 30 nm, which indicates that the agglomerates in the powder consist of 2–3 crystallites (Figure 7).

Figure 8 shows the TEM data for the composite synthesized from the zirconia calcined at 500 °C and a suspension of the graphene in isopropanol. Two types of particles are clearly distinguishable: sheets of graphene, translucent, flat or twisted, and more optically dense agglomerates with sizes of less than 100 nm, consisting of zirconia nanocrystals (Figure 8a). The electron diffraction in the sample area shown (inset), is the result of a superposition of a group of reflections corresponding to the ZrO_2_ lattice and reflections due to the multiple graphene sheets of low layering and shifted, relative to each other at different angles. An enlarged fragment of Figure 8a is shown in Figure 8b, where the ZrO_2_ nanocrystals appear attached to the surface of the graphene sheets. The automatic measurement of the crystallites shows that their size does not exceed 10 nm. In the dark-field image in Figure 8c, the moiré pattern indicates that the graphene particles were formed from 1–2-layer sheets; it can be seen that the zirconia nanocrystals agglomerate is fixed over the moiré pattern. The dark-field image in Figure 8d, clearly shows that the zirconia crystallites on the graphene are distributed discretely. According to the elemental analysis, the carbon content in the composite was 0.635 (±0.007) wt%.

A comparative study of the effect of adding graphene on the compaction dynamics, during the pressing of nano-zirconia was carried out, and Figure 9 shows the typical deformation–pressure curves, which have been drawn, based on the results of five or six independent numerical experiments and kinetic studies of the pressure dependence. The ultimate strain values corresponding to the deviations from the linear compression of the powders are also noted. The powder compaction curves consist of three sections: linear, parabolic, and exponential. At the initial stage of the compaction, at low pressures, the linear section of the curve corresponds to the elastic deformation, while the stress-strain curve corresponds to Hooke’s Law. The length of the linear part of the curve for the composite is much shorter than that for the single, initial, nano-zirconia. Since the tangent of the angle of the tangent for the initial section is equal to the compressibility modulus (k_c_) of the powder, Figure 9 shows that the k_c_ of the nanostructured hybrid is several times higher than that of the single nano-zirconia. We have seen a similar pattern earlier [33], and this study confirms the fact that the graphene sheets in a hybrid nanostructure make it difficult to press, despite the fact that the graphene was introduced into the nano-zirconia in different ways.

It is probable that the reason for the observed phenomenon, is the elasticity of the graphene sheets, which prevents the system from being compressed at the stage of the elastic deformation. As a result, the composite passes into the plastic deformation region at a lower pressure than the single nano-zirconia. It can also be seen that a higher pressure is required to achieve the same degree of deformation for the single nano-zirconia. It can also be assumed that, at the stage of the elastic deformation, due to the van der Waals interaction of the hybrid nanoparticles, the particles of a more regular shape, close to isometric, are formed because by optimizing their arrangement in the volume.

Next, the parabolic sections of the curves correspond to the transition of the systems to the stage of accommodation (adaptation), where the particles begin to move relative to each other to more effectively fill the voids between the larger particles with smaller ones. At this stage, only an insignificant contribution of the particle deformation to the process of the compaction of powders was observed [41].

The final, exponential sections of the shrinkage curves correspond to an intense increase in the stress with small changes in the deformation, apparently because there is practically no free space left in the compact, and only the particles themselves can be deformed. A comparison of the experimental data for the graphene-containing composite and the initial single nano-zirconia, shows that the compressibility modulus of the latter is much lower, while it should be noted that a higher pressure is required to achieve the same degree of deformation for single nano-zirconia.

A comparative study of the shrinkage of the nanostructured graphene-zirconia and the nano-zirconia powders, was carried out. Figure 10 shows the thermal curves of shrinkage and the shrinkage rates for the nanostructured graphene-zirconia composite and the single zirconia nanopowders. A curve analysis in Figure 10a shows that the onset of the sintering, preceded by some thermal expansion (by 1.2%), is observed at 615 °C. The reduction in the sample size by ~10%, occurs almost uniformly in the temperature range of 614–1178 °C with a slowdown in the region of 900–1178 °C.

In the temperature range of 1170–1178 °C, a break is observed on the dL/L_0_ curve with a sharp increase in the shrinkage rate (dL/dt) from 0.42%/min (1210 °C) to the maximum value, 0.85%/min (1395 °C), in the region of 1400 °C. The observed break in the curve coincides with the temperature of the ZrO_2_ phase transition from the monoclinic to the tetragonal modification (1170 °C). The total shrinkage of the sample over the entire temperature range studied (from room temperature to 1600 °C), was more than 24%.

For comparison, Figure 10b shows the results of a dilatometry study under the same conditions of a calcined single ZrO_2_ nanopowder, which was used to obtain a composite with graphene. The initial thermal expansion of the sample was 2.06%. The start of the sintering was recorded at the temperature of 595.8 °C, and it corresponded to a sharp increase in the shrinkage rate of the sample, which exceeded 0.26%/min at 681 °C. The observed temperature of the phase transition from the monoclinic to the tetragonal modification was 1165–1166 °C, when the sample shrinkage (dL/L_0_) did not exceed 0.5%. Then, the accelerated growth of the shrinkage was observed at the final stage of the experiment, in the temperature range 1439–1600 °C. The maximum speed, in this case, was more than 0.38%/min at 1561 °C, and the total shrinkage of the sample in the entire experiment was more than 9%.

For comparison, the results of a dilatometric study of the synthesized graphene-containing composite and a single nano-zirconia, are presented in the summary Table 1. Its analysis shows that the dynamics of the sintering of the composites, based on both the graphene and nano-zirconia, differs from the dynamics of the sintering of the single nano-zirconia, apparently due to the presence of the graphene sheets in the structure of the powder starting material, which affects the shrinkage rate, the total shrinkage, and the final geometric dimensions of the sample. At the initial stage of the sintering, the value of the thermal expansion of the composite was less than 60% of this indicator for a single nano-zirconia. At the same time, the beginning of the composite sintering was recorded at a temperature of 614 °C, i.e., 19 °C higher than in the case of the single nano-zirconia. The most pronounced effect of the graphene in the structure was the affected shrinkage rate and the total shrinkage of the samples. The composite showed the maximum shrinkage, which exceeded this indicator for a single nano-zirconia by 2.6 times.

The effects found can only be explained by the presence of the graphene sheets, which, firstly, restrain the expansion of the zirconia agglomerates and, secondly, partially prevent the disordering of the surface monolayers of the zirconia crystallites, prior to the sintering. Apparently, the graphene sheets “enveloping” the zirconia crystallites partially reduce the surface diffusion rate of the atoms forming these crystallites at the initial and middle stages of the sintering in the temperature range up to 1000 °C. At the same time, in the region of 650–700 °C, the shrinkage is approximately 8 and 7% of the initial sample sizes for the composite and the single nano-zirconia, respectively.

The effect of the graphene sheets on the rate and magnitude of the shrinkage is most pronounced at temperatures above 1000 °C. An increase in the shrinkage rate and the total shrinkage of the composite, compared to a single nano-zirconia, may be due to an increase in the surface energy of the zirconia crystallites, as a result of the van der Waals interaction with the sp^2^-electron system of the graphene (δ−). Figure 11 schematically illustrates this phenomenon due to the numerous defects, usually the oxygen vacancies (δ+), on the surface of the ZrO_2_ nanocrystals. The increased defectiveness of the crystal structure is known to be typical of nano-objects, which is confirmed by the XRD data for the single nano-zirconia synthesized, which was subsequently used to obtain a composite with graphene.

As shown above, based on the analysis of the dilatometric data, as well as the results of the kinetic calculations, the effect of the addition of 0.635 wt% graphene on the sintering dynamics of the nanostructured zirconia powder and the mass transfer mechanism during the sintering was shown. According to the results of the calculation presented in Table 2, the value of the activation energy of the sintering of the single nano-zirconia was 176 kJ/mol (Appendix A), while for the composite, this value was 382 kJ/mol (Appendix A), i.e., more than two times higher. Apparently, the graphene partially stabilizes the zirconia structure and prevents the disordering of the surface monolayers of the nanocrystals and their melting, prior to the sintering, which, as a result, leads to an increase in the activation energy of this process for the composite. As the analysis of the obtained data shows, in the presence of graphene, the mechanism of mass transfer during the sintering changes. In the presence of the graphene sheets in the hybrid structure, its sintering occurs through the stage of the grain boundary diffusion, in contrast to the sintering of the single nano-zirconia through the viscous flow.

Figure 12 shows the hypothetical sintering mechanisms for a single nano-zirconia and a graphene-containing composite based on it, compiled from our results. When considering these mechanisms, we proceeded from the following fundamental physicochemical concepts: the surface of the nanostructured powder particles is saturated with pores and defects (i), the nanocrystals in the agglomerates are multidirectional (ii), the particles of the nanocrystalline powders have a high excess free energy (iii), the diffusion is the main promoter of the sintering process (iv), the system, as a whole, tends to reduce the free surface area (v).

With an increase in temperature, the disordering occurs in the surface monolayers of the zirconia particles, especially on the edges and tops of the crystallites, as well as on the protrusions of the agglomerates, i.e., in the regions of higher surface energy. At the initial stage, at a relatively low activation energy and a large surface area of the particles, the viscous flow dominates, due to the high shear stresses in the contact area.

The beginning of the sintering is accompanied by an increase in the contact area between the particles and their initial association with each other; in this case, the particles retain their structural individuality, and their contours are practically preserved. Subsequently, because of the mass transfer, an exchange occurs between the vacancies and the highly mobile atoms, and in the interparticle spaces, the latter are concentrated and have become shared by several particles. This leads to the gradual settling of vacancies, the filling of pores and compaction. With an increase in temperature, the bulk diffusion becomes more intense, and with the strengthening of the interparticle bonds, the diffusion at the grain boundaries becomes more active.

In the case of a composite powder, a change in the sintering mechanism is observed: instead of the viscous flow process, a mixed mechanism is realized in the contact areas of the zirconia particles, in which the grain boundary diffusion prevails. In this case, the second component of the mechanism is the volume diffusion in those areas where the clean surfaces of the zirconia nanoparticles interact with the subsequent coalescence and the formation of the uniform grain boundaries. Where the graphene sheets are between the zirconia particles, most likely, the grain-boundary diffusion occurs with the movement of the surface ZrO_2_ monolayers parallel to the graphene sheets, resulting in a change in volume and shape and the compaction of the ceramic particles [42]. The combination of the grain boundary and the bulk diffusion results in a reduction in the content of defects, which, in turn, also contributes to the compaction of the sintered material.

## 4. Conclusions

The effect of the graphene sheets on the rheological and sintering properties of a nanostructured composite powder, based on the zirconia, has been studied. The work covers the research from the synthesis of the nano-zirconia powder to a graphene-containing nanostructured composite, based on it. The paper reports on a method for synthesizing a nanostructured composite, using two suspensions: the nano-zirconia obtained by the sol-gel synthesis and the oxygen-free graphene obtained by a sonochemical method. The morphology of the oxygen-free graphene sheets, the phase composition and the morphology of the zirconia powder, as well as the morphology of the synthesized composite were studied. A comparative study of the rheology, the dynamics of the sintering, and the activation energy of the sintering of the pure nano-zirconia and a graphene-containing composite based on it, has been carried out.

It has been found that the graphene sheets in a hybrid nanostructure make it difficult to press at the stage of the elastic deformation, and the compressibility modulus of the nanostructured hybrid is several times higher than that of the pure nano-zirconia. As a result, the composite goes into the region of the plastic deformation at a lower pressure than the single nano-zirconia. A mechanism for sintering a composite with a graphene content of 0.635 wt% is proposed; it explains the effect of the graphene sheets on the process and its result. It has been determined that the activation energy of the composite sintering is more than two times higher than for the single nano-zirconia. Apparently, due to the van der Waals interaction, the graphene sheets partially stabilize the zirconia structure. They inhibit the disordering of the surface zirconia monolayers and their sub-melting, the two processes preceding the sintering. The determined effects lead to an increase in the activation energy of the sintering for the composite and a complication in its mechanism. As the analysis of the obtained data shows, in the presence of the graphene sheets in the hybrid structure, its sintering occurs according to a mixed mechanism, in which the grain boundary diffusion predominates, in contrast to the sintering of the pure nano-zirconia, which proceeds through a viscous flow stage. The study of the physicochemical properties of the graphene-containing composites shows that the oxygen-free graphene sheets impart the rheological and morphological features to the composite powders based on the nano-zirconia, and also increases its sintering activation energy. The determined patterns open up new capabilities for the modeling of the compacting and sintering processes when creating new types of fine-grained ceramics for the electrodes of lithium-ion batteries, sensors, details of small-sized devices, and lightweight protective coatings.

## Figures and Tables

**Figure 1 materials-15-07342-f001:**
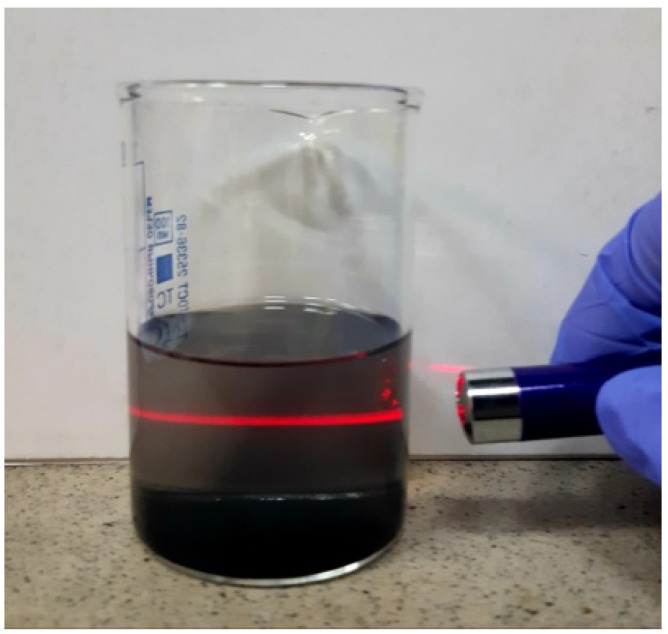
Tyndall effect in the graphene suspension in isopropanol after 24 h exfoliation.

**Figure 2 materials-15-07342-f002:**
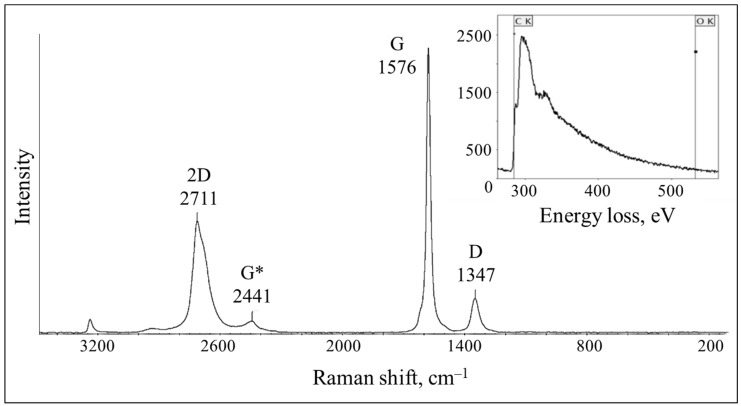
Raman and EELS (inset) spectra of the graphene obtained in isopropanol by ultrasonic treatment. G*—overtone.

**Figure 3 materials-15-07342-f003:**
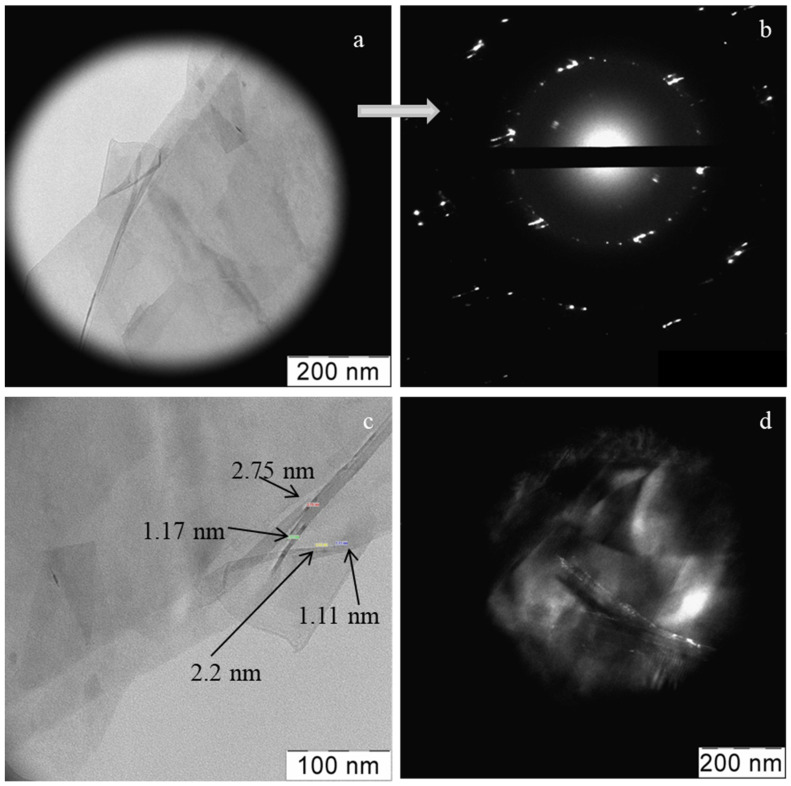
TEM data for a graphene suspension in isopropanol: a package consisting of sheets (**a**); electron diffraction in the area shown in subfigure (**a**,**b**); the double thickness of the graphene sheets on the folds (**c**), the dark-field image shows the moiré pattern on the graphene (**d**).

**Figure 4 materials-15-07342-f004:**
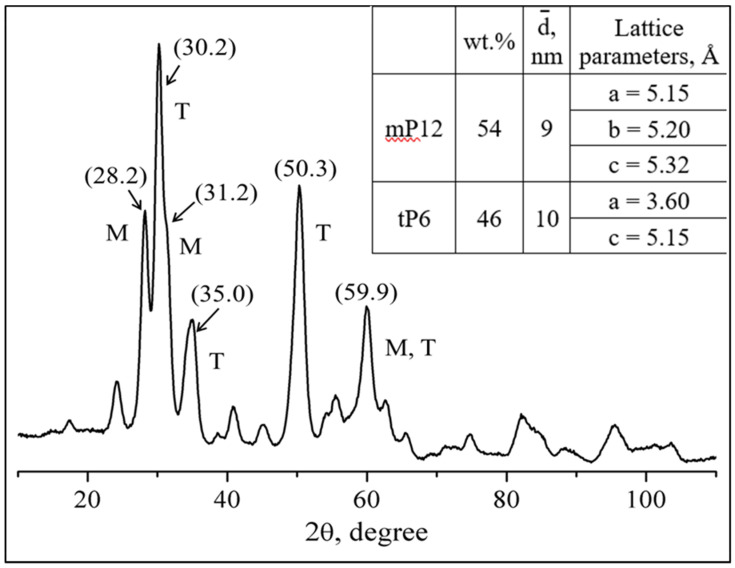
XRD pattern and phase composition, as well as the crystallographic data for the zirconia nanopowder (inset).

**Figure 5 materials-15-07342-f005:**
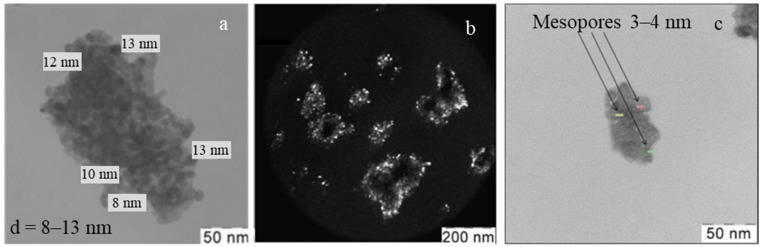
TEM data for the nano-zirconia synthesized: light-field (**a**,**c**) and dark-field (**b**) images.

**Figure 6 materials-15-07342-f006:**
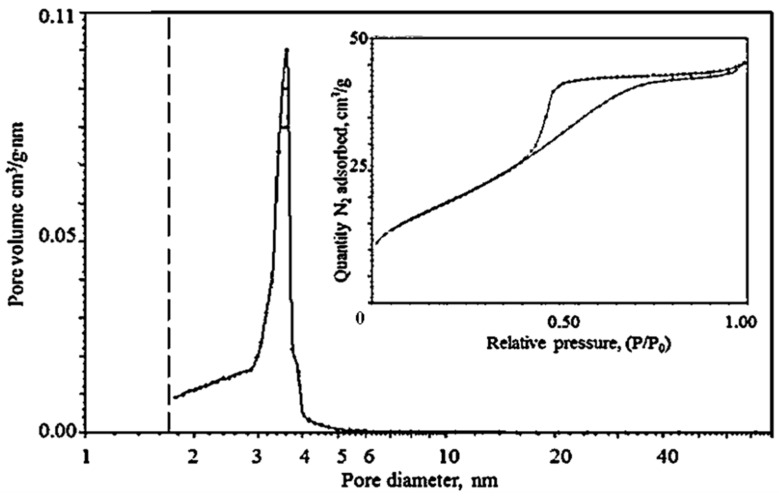
N_2_ adsorption–desorption data for the nano-zirconia powder: pore size distribution and adsorption isotherms (inset).

**Figure 7 materials-15-07342-f007:**
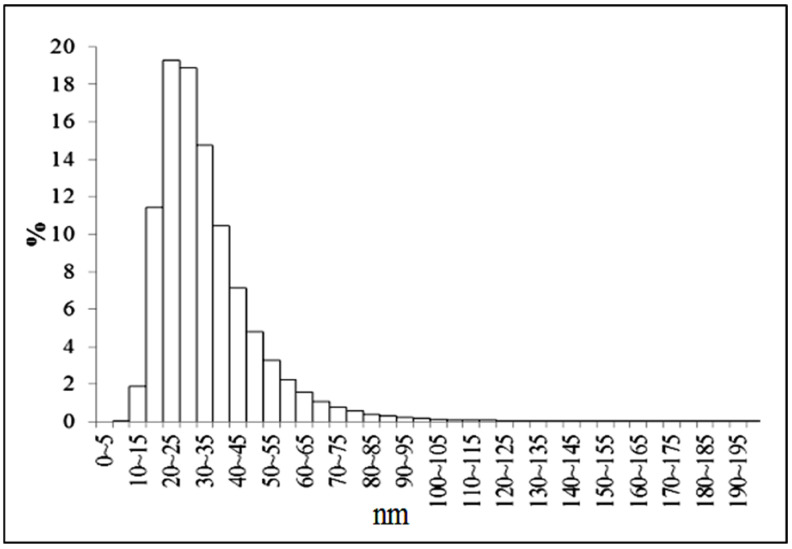
Particle size distribution for the zirconia nanopowder.

**Figure 8 materials-15-07342-f008:**
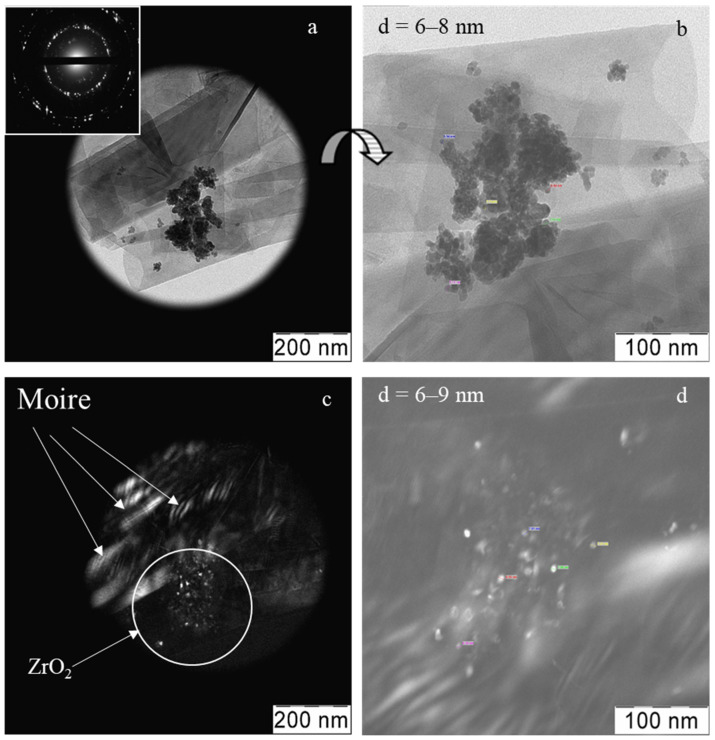
TEM images for the graphene-zirconia composite synthesized from the nano-zirconia and the graphene suspensions in isopropanol: light-field image and electron diffraction (inset) (**a**), enlarged and rotated by the 180° light-field image of the same area (**b**), dark-field image of the ZrO_2_ agglomerate (marked by circle) between graphene sheets (moiré) (**c**), enlarged fragment with the automatically measured crystallites sizes (**d**).

**Figure 9 materials-15-07342-f009:**
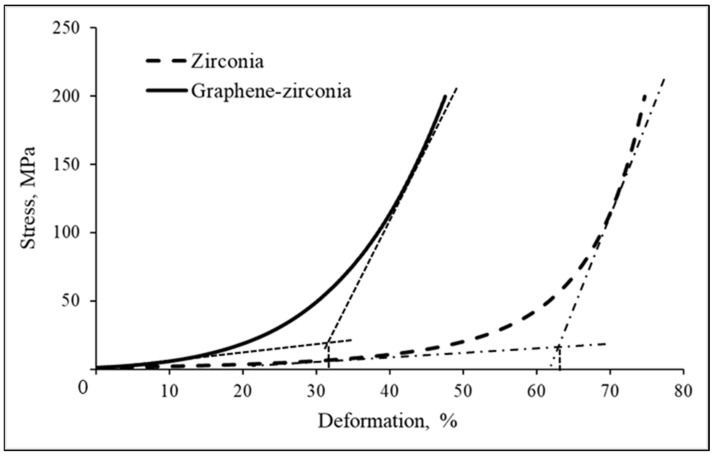
Compaction curves on the deformation–pressure plane of the graphene-zirconia and single zirconia nanostructured powders.

**Figure 10 materials-15-07342-f010:**
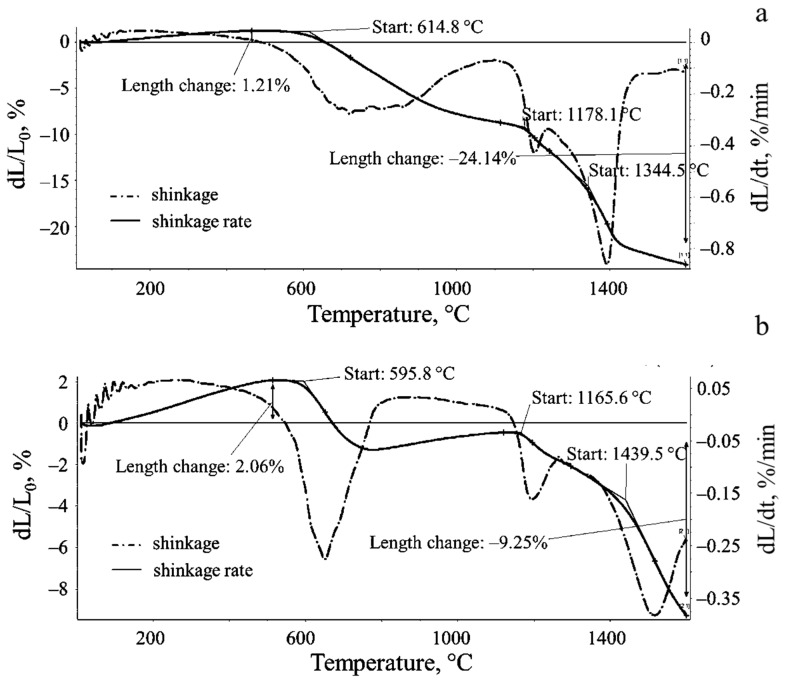
The shrink and shrink rate curves for the graphene-zirconia composite (**a**) and the single nano-zirconia (**b**).

**Figure 11 materials-15-07342-f011:**
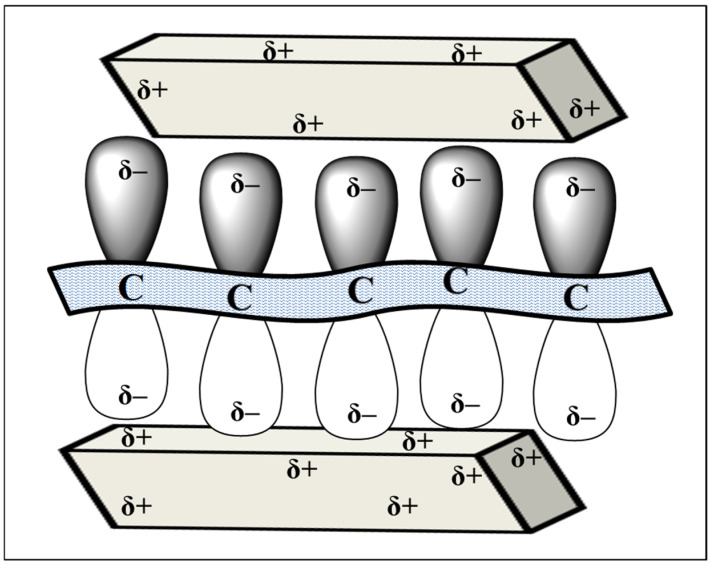
Schematic image of the van der Waals interaction between the graphene sheets and the zirconia nanocrystals in the composite nanopowder.

**Figure 12 materials-15-07342-f012:**
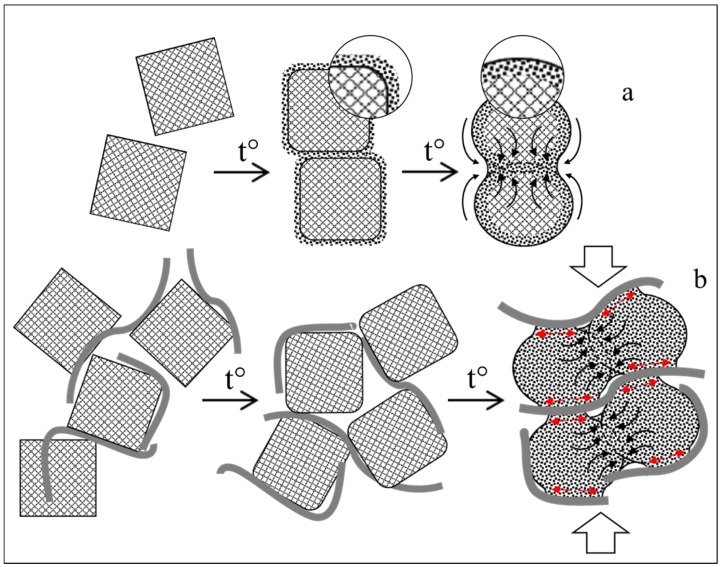
Schemes of the possible sintering mechanisms of the single nano-zirconia (**a**) and the graphene-zirconia composite (**b**).

**Table 1 materials-15-07342-t001:** Results of a dilatometric study of the graphene-zirconia composite and the nano-zirconia used in its synthesis.

	Thermal Expansion at the Initial Stage, %	Starting Temperature of Sintering, °C	Phase Transition Temperature, °C	Full Shrinkage, %	Maximum Shrinkage Rate, %/min
Graphene-zirconia	1.21	614	1178	24.00	0.85
Nano-zirconia	2.06	595	1165	9.25	0.38

**Table 2 materials-15-07342-t002:** Parameters and mechanisms of the sintering of the nanostructured zirconia and the graphene-containing composite based on it.

Powder	Reaction Order, n	Sintering Activation Energy, Q, kJ/mol	Sintering Mechanism
Nano-zirconia	1.87	176	Viscous flow
Graphene-zirconia	0.39	382	Mixed mechanism with a grain boundary diffusion predominance

## Data Availability

Not applicable.

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
