# Peer review of "Influence of Graphene Sheets on Compaction and Sintering Properties of Nano-Zirconia Ceramics"

_materials, 2022, doi:10.3390/ma15207342_

Round 1

Reviewer 1 Report

In this work, Trusova et. al., synthesized nanozirconia-graphene composite and investigated the effect of graphene sheets on the rheological and sintering properties. The characterization and result and discussion parts are well described. However, there are some comments should be addressed before acceptance.

 1.    Line 70 and line 79 and so on, please add the author’s name for article [16], [21], ,In [22], publication [23], publication [24], the work [28].

2.    Please add detail of Figure 5 (a) (b) (c) in line 291 and Figure 8 (a), (b), (c), (d) in line 342.

3. The author cite reference 31 as your earlier publication, however, in reference 31, the author’s name in reference 31 and in this manuscript were not the? Please check reference 31 again.

           4.  Please correct some typos and grammatical error in this manuscript.

Author Response

Dear Colleague

Thank you very much for your attention to our manuscript. We have used your comments to improve the quality of the text. The names of the first citing authors have been included in the text, which has been edited accordingly.

The details of Figs 5 and 8 were supplemented and edited, and we also checked the other figures and clarified the captions.

The annoying shift of references in the reference section has been eliminated, and all referenceshave taken their proper places, now their numbering corresponds to the text.

We have tried to correct typos and grammatical errors as completely as possible.

With thanks and best wishes

  1. Trusova
  2. Titov
  3. Afzal
  4. Abramchuk

Reviewer 2 Report

Please see the enclosed file.

Author Response

Dear Colleague

Thank you very much for your attention to our manuscript. We have used your comments to improve the quality of the text. We tried to overcome the vagueness of some sentence and checked the entire text for the presence of such «nebulae». Indeed, in publications about new compositions, authors often want to offer new material in the hope that it will be in demand by engineers. We plan to study the electro-physical characteristics of our compositions in the coming days and will devote the following publications to them.

The phrases you noted have been rewritten in accordance with your recommendations.

The drawing with RAMAN has been redone, the spelling of the units has been corrected.

With thanks and best wishes

  1. Trusova
  2. Titov
  3. Afzal
  4. Abramchuk

Reviewer 3 Report

This paper, Influence of graphene sheets on compaction and sintering properties of nano-zirconia ceramics (Materials- 1933863), prepared a composite using two suspensions: nano-zirconia obtained by sol-gel synthesis and oxygen-free graphene obtained sonochemically. It was found that graphene is an important factor affecting the process mechanism, the activation energy of composite sintering is more than two times higher than for single nano-zirconia. And here are some comments on this paper, which may help improve the quality of this work.

1.      The potential application of the obtained hybrid composite power should be well described to highlight the value of this work.

2.      The property of the obtained hybrid composite should be tested to highlight the potential value of this work.

3.      The diffraction peak in the XRD pattern should be indexed and the corresponding number of the index pdf card should be displayed.

4.      Why were the Rheological measurements and dilatometric conducted for the composite powders?

5.      Following the last question, why not the coefficient of expansion as not displayed but the dilatometric?

6.      It is well known that the dilatometric property is intensively dependent on the density of the material, and the dilatometric test requires a solid bulk sample, how to make such a sample diameter of 5 mm and a height ∼of 2.5 mm?

7.      It is clear that the graphene aggregated at the bottom of the cup in figure 1, how to solve the dispersion of graphene solution? Here is a related work for improving the dispersion uniformity o graphene solution is recommended (Composites Part B 216 (2021) 108832).

8.      References published in recent five years should be cited to enhance the timeliness of this work.

Author Response

Dear colleague

Many thanks to the respected reviewer for a detailed analysis of the shortcomings of our manuscript. We have tried to strengthen the indications of the high applied potential of the developed compositions, while maintaining, however, rationality and modesty.

As is known, in terms of electronic and other physicochemical properties, oxygen-free graphene is fundamentally different from its modified forms, oxidized and reduced oxidized with a destroyed sp2system. The determined features of the physicochemical properties of composite nanosystems indicate the role of graphene, which is an original part of the presented work. In the text, we have emphasized the role of oxygen-free graphene sheets in the formation of the structure and physicochemical properties of the powder nanosystems, which dictate their own requirements for compaction and sintering modes. The latter require special studies, which will be the subject of our subsequent work.

The figure with the diffraction pattern has been replaced, the card numbers are contained in the text.

The suspension of oxygen-free graphene obtained by the sonochemical method is resistant to delamination, and there is no vertical density gradient in it. This property is ensured by the method of preparation described in paragraph 2.1. Stepwise treatment of the as-prepared graphene colloid (after ultrasonic exfoliation), which consists in successive sedimentation and decanting, makes it possible to obtain a stable graphene suspension, as reported in a number of our articles [Trusova, E.A. et al. RSC New J. Chem. (2021) 45(23), 10448-10458; Fuller. Nanotubes & Carbon Nanostruct. (2021) 29(6), 431-441; RSC Nanoscale Adv. (2020) 2(1), 182-189; Diam. Rel. Mater. (2018) 85, 23-36, Adv. Mater. Scie. Eng. (2018) 2018, 6026437 (11p)]

We supplemented the References with publications from recent years, after which the share of publications from the last 5 years is more than 40%.

We read with interest an article recommended by a respected reviewer, which is devoted to the study of the strengthening effect of reduced oxidized graphene on copper matrix composite materials. It is close to our understanding of the processes that occur during the interaction of graphene and metal-containing colloidal particles, as a result of which hybrid nanostructures are formed. Thanks a lot. We use this publication as a reference in the preparation of the following manuscript.

The study of the rheological and dilatometric properties of a composite with graphene is especially important for studying the nature of the influence of graphene on oxide ceramics. Our previous studies have shown that even a small amount of additive dramatically affects the compressibility of the powder. To study the activation energy of powder sintering, it is important that all sample components have the closest possible relative densities.

You are absolutely right about the relationship between initial density and dilatometric properties. It is for this reason that rheological studies of the compressibility of the powder were previously carried out in order to use a green body of the same density in both cases. Due to the low yield of the initial powder with graphene and low bulk density, it is difficult to prepare a series of larger samples (than 5 mm in diameter and ∼2.5 mm in height). However, I want to emphasize that the green body with graphene and pure nano-zirconium dioxide after cold isostatic pressing is quite strong. We have the technical means to produce tablets of this size. This makes it possible to use such miniature samples for dilatometric studies of the sintering process.

The cooling curves were used to determine the CTE of the sample. Since these curves were linear without any additional important information, we did not overload the manuscript with them. However, to calculate the activation energy, these values were used in formula (1).

With thanks and best wishes

  1. Trusova
  2. Titov
  3. Afzal
  4. Abramchuk

Round 2

Reviewer 2 Report

Acceptable

Author Response

Thank you!

Reviewer 3 Report

The response to comments should be basically corresponded to each question, and give a direct, correct, and responsible answer to the question. For example, the potential application of the materials prepared in this work may be very familiar to the authors who is focusing on studying it, but it does not mean that readers in other fields knows it. Consequently, it is very importance to highlight the potential value and the application background of the materials. If not, the value or novelty of this work are severely impacted. Another important issue is the necessity or potential value of carrying out these structure and property tests. There should be a proper description on the aim of the structure characterization or property tests to lead readers to think in the right way. Finally, we do not talk on other irrelevant issue such as the next paper, it should be  kept on the current work.

Author Response

Dear Reviewer,

Many thanks to the respected reviewer for a detailed analysis of the shortcomings of our manuscript. We tried to take into account all your comments.
  1. The potential application of the obtained hybrid composite power should be well described to highlight the value of this work.

As is known, in terms of electronic and other physicochemical properties, oxygen-free graphene is fundamentally different from its modified forms, oxidized and reduced oxidized with a destroyed sp2 system. The determined features of the physicochemical properties of composite nanosystems indicate the role of graphene, which is an original part of the presented work. In the text (Lines 115-133), we have emphasized the role of oxygen-free graphene sheets in the formation of the structure and physicochemical properties of the powder nanosystems, which dictate their own requirements for compaction and sintering modes.

  1. The property of the obtained hybrid composite should be tested to highlight the potential value of this work.

Lines 141-145: “We turned to rheological studies, because the behavior of the systems we develop under standard conditions differs significantly from the behavior of traditional submicron powders due to the presence of ZrO2nanocrystals and graphene sheets in the material, which significantly change the fluidity and compressibility of the powders. At the same time, there are practically no works covering these physicochemical characteristics.”

  1. The diffraction peak in the XRD pattern should be indexed and the corresponding number of the index pdf card should be displayed.

Lines 307-308: (JCPDS card no. 24-1164) (JCPDS card no. 05-0543).

  1. Why were the Rheological measurements and dilatometric conducted for the composite powders?

Lines 141-145: “We turned to rheological studies, because the behavior of the systems we develop under standard conditions differs significantly from the behavior of traditional submicron powders…”

The study of the rheological and dilatometric properties of a composite with graphene is significant for studying the nature of the influence of graphene on oxide ceramics. Our previous studies have shown that even a small amount of additive dramatically affects the compressibility of the powder.

For a more accurate calculation of the sintering activation energy of powder, it is important that all sample components have the closest possible relative densities.

  1. Following the last question, why not the coefficient of expansion as not displayed but the dilatometric?

Lines 242-245: The cooling curves were used to determine the CTE of the sample. Since these curves were linear without any additional important information, we did not overload the manuscript with them. However, to calculate the activation energy, these values were used in formula (1)

  1. It is well known that the dilatometric property is intensively dependent on the density of the material, and the dilatometric test requires a solid bulk sample, how to make such a sample diameter of 5 mm and a height of 2.5 mm?

You are absolutely right about the relationship between initial density and dilatometric properties. It is for this reason that rheological studies of the compressibility of the powder were previously carried out in order to use a green body of the same density in both cases. Due to the low yield of the initial powder with graphene and low bulk density, it is difficult to prepare a series of larger samples (than 5 mm in diameter and ∼2.5 mm in height). However, I want to emphasize that the green body with graphene and pure nano-zirconium dioxide after cold isostatic pressing is quite strong. We have the technical means to produce tablets of this size. This makes it possible to use such miniature samples for dilatometric studies of the sintering process.

  1. It is clear that the graphene aggregated at the bottom of the cup in figure 1, how to solve the dispersion of graphene solution? Here is a related work for improving the dispersion uniformity o graphene solution is recommended (Composites Part B 216 (2021) 108832).

The suspension of oxygen-free graphene obtained by the sonochemical method is resistant to delamination, and there is no vertical density gradient. This property is ensured by the method of preparation described in paragraph 2.1. Stepwise treatment of the as-prepared graphene colloid (after ultrasonic exfoliation), which consists in successive sedimentation and decanting, makes it possible to obtain a stable graphene suspension, as reported in a number of our articles [Trusova, E.A. et al. RSC New J. Chem. (2021) 45(23), 10448-10458; Fuller. Nanotubes & Carbon Nanostruct. (2021) 29(6), 431-441; RSC Nanoscale Adv. (2020) 2(1), 182-189; Diam. Rel. Mater. (2018) 85, 23-36, Adv. Mater. Scie. Eng. (2018) 2018, 6026437 (11p)]

  1. References published in recent five years should be cited to enhance the timeliness of this work.

Lines 627-652: We supplemented the References with publications from recent years, after which the share of publications from the last 5 years is more than 40%.

With thanks and best wishes

  1. Trusova
  2. Titov
  3. Afzal
  4. Abramchuk